# Metagenomic Sequencing to Analyze Composition and Function of Top-Gray Chalky Grain Microorganisms from Hybrid Rice Seeds

**DOI:** 10.3390/plants12122358

**Published:** 2023-06-18

**Authors:** You Liu, Yuan Yuan, Hui Yuan, Yan Wang, Chenzhong Jin, Hao Zhang, Jianliang Tang, Yihong Hu

**Affiliations:** 1College of Agriculture and Biotechnology, Hunan University of Humanities, Science and Technology, Loudi 417000, China; 2Quality Inspection Department, Hunan Ava Seeds Co., Ltd., Changsha 410128, China; 3Quality Inspection Department, Baiyunshan Forest Farm of Shimen County, Changde 415312, China

**Keywords:** top-gray chalkiness, hybrid rice seeds, metagenomics sequencing, glycoside hydrolases, fungi, microorganisms

## Abstract

The top-gray chalkiness of hybrid rice (*Oryza sativa* L.) seeds is a typical phenomenon in hybrid rice seeds. The chalky part of the grain is infected and is the inoculum to infect the normal seeds during storage and soaking. These seed-associated microorganisms were cultivated and sequenced using metagenomics shotgun sequencing to obtain more comprehensive information on the seed-associated microorganisms in this experiment. The results showed that fungi could grow well on the rice flour medium, similar to the ingredients of rice seed endosperms. After the assembly of metagenomic data, a gene catalog was established, comprising 250,918 genes. Function analysis showed that glycoside hydrolases were the dominant enzymes, and the genus *Rhizopus* accounted for the dominant microorganisms. The fungal species *R. microspores*, *R. delemar*, and *R. oryzae* were likely to be the candidate pathogens in the top-gray chalky grains of hybrid rice seeds. These results will provide a reference for improving hybrid rice processing after harvest.

## 1. Introduction

Hybrid rice (*Oryza sativa* L.) is one of the staple foods for people in the world, and China is a major producing area for hybrid rice. Since the 1970s, seed production technology in relation to hybrid rice has developed rapidly [1]. Different from the conventional rice varieties, hybrid rice seeds are prone to more serious glume dehiscence owing to their male sterile line, long sunny light, as well as warm environmental temperatures during the pustulation period, often leading to heavy degradation in terms of seed quality during field production [2,3]. This glume dehiscence phenomenon after anthesis results in a declining germination rate and bud potential, causes rice kernel smut and preharvest sprouting in hybrid rice seeds, and ultimately reduces seed yield and quality [4,5]. In addition, these “glume-opening” seeds are susceptible to infection, thus reducing seed activity during transportation and storage, especially in high temperature and humidity weather, and even polluting healthy seeds during seed soaking before field production [6].

Hainan Province, China, is located between 18°10′–20°10′ N and 108°37′–111°3′ E, where it is rainy and warm all year round, and this area is an ideal area to quickly produce hybrid rice seeds, especially for winter breeding [7]. In recent years, high temperature and humidity have become negative issues, impacting hybrid rice breeding in this area. We previously observed that hybrid rice seeds exhibited quite a high proportion of top-gray chalkiness, in which the chalky part originated from the upper part of the grain opposite to the embryo other than the five types of chalkiness reported by Yoshioka et al., and it was noticed that the chalkiness rate of seeds was closely related to the glume dehiscence rate, where the higher the ambient temperature and humidity were, the higher the chalkiness rate emerged [8,9]. This phenomenon has been seriously troubling the processing of hybrid rice seeds.

In the process of seed processing in the packing workshop, there are two ways to overcome the consequences of top-gray chalkiness in hybrid rice seeds: one approach is called ‘light selection technology’ to identify and pick out the chalky grains in the processing line in real-time according to the seed digital images acquired by using high-speed photography illuminated from behind, in which light passes through the tissue of the rice grain [10,11]; the other is to use seed-coating agents to repair and disinfect the glume-dehiscent rice seeds [12]. Undoubtedly, the application of seed-coating agents is a low-cost method in seed processing. The key issue of applying this technology is to figure out the compositions of harmful microorganisms in these defective seeds.

Our previous research showed that the starch granules of the chalky parts of hybrid seeds collapsed and were infected, which could infect normal seeds during seed storage, processing, and soaking periods, and seed coating agents containing fungicides or bactericides could suppress this phenomenon to an extent [8,12]. The seed-associated microorganisms are quite complicated, and there are still few papers on the transmission and infection mechanism of top-gray chalkiness in hybrid rice seeds [13]. Our previous work has targeted the 16S rRNA and ITS amplicon sequencing to describe the chalky grain microorganisms, but the acquired information was limited. In this study, metagenomic shotgun sequencing was conducted with the microorganisms from the top-gray chalky seeds to obtain more comprehensive and detailed information.

The main purpose of this study is to obtain a better understanding of the microbiota of chalky rice grain to facilitate the application of appropriate seed coating agents in seed processing in large-scale industries. These research results will play a guiding role in improving hybrid rice seed processing after harvest.

## 2. Results

### 2.1. Cultivated Microorganisms from Chalky Rice Seeds

From the chalky rice seeds without husks, microorganisms were cultivated on beef extract peptone medium (TA) and rice flour medium (TB), respectively. Only bacterial plaques were observed on the beef peptone medium (Figure 1A), whereas hyphae were observed on the rice flour medium (Figure 1B). Then, the TB group was sub-cultured on the same medium after steak (TBS), and only hyphae and spores were observed (Figure 1C,D), which conformed to the typical fungal morphological characteristics. These results indicated that the rice flour medium was available to cultivate fungi from the chalky rice seeds.

### 2.2. Microbiome Diversities of Chalky Rice Seeds Cultivated on Different Media

A total of 55,870.57 Mbp raw data were acquired after sequencing, and then an amount of 55,824.05 Mbp clean data were acquired. After quality control to eliminate the sequences of host contamination, an amount of 55,725.44 Mbp non-host data were acquired (Appendix A). The effective rate of the data was 99.82%. This indicated that the acquired data were effective for further metagenomic assembly. After assembly, a total number of 570,029 open reading frames (ORFs) were predicted, the gene catalog was established, comprising 250,918 genes, and the integrated gene catalog (IGC) was established, comprising 149,383 genes (Appendix A).

As shown in Figure 2A, the gene number in the TBS group was reduced to a greater extent than that in the TB group. A Venn diagram showed that 35,365 genes co-existed in all groups, and 72,220 genes co-existed only in the TB group and TBS group (Figure 2B). The Krona method directly showed that bacteria were dominant in TA (Figure 2C). As shown in Figure 2D, at the phylum level, the phyla Proteobacteria and Firmicutes were dominant in the TA group, and the phylum Mucoromycotya was dominant both in the TB group and TBS group; at the species level, the species *Rhizopus delemar* and *R. microspores* were the most abundant species in the TB group and TBS group, respectively. As shown in Figure 2E, *R. delemar* was more abundant in the TB group, and the genus *Rhizopus* and the species *R. oryzae* were also more abundant in the TB group. While in the TA group, the bacterium genera *Acinetobacter* and *Lysinibacillus* were the most abundant. Moreover, the family Mucoraceae was more abundant in the TB group (Figure 2F). The heatmap based on LEfSe showed that the genus *Rhizopus* was dominant both in the TB group and the TBS group (Figure 2G). Metastats analysis also proved that *R. oryzae* was more abundant than the others (Figure 2H,I). Taken together, these results revealed that only the bacteria from the chalky grains could be cultivated on the beef extract peptone medium, whereas the fungi from the chalky grains could be cultivated well on the rice flour media that were similar to rice endosperm composition. Both in the TB group and TBS group, the fungus genus *Rhizopus* was dominant, further indicating that it could be a possible deteriorating factor during seed germination.

### 2.3. Functional Analysis of Microorganisms from Chalky Rice Seeds Cultivated on Different Media

Metagenomic data can also be applied for functional analysis conveniently. To compare the functional genes among the groups, we chose the professional database for carbohydrate enzymes CAZy in our study. Among the established gene catalog of 250,918 genes (Appendix A), a total of 6281 genes (2.50%) were compared to the CAZy database.

As shown in Figure 3A, the most abundant function category was glycoside hydrolases (GHs). The gene clusters of samples at level 1 showed that the TB group and the TBS group had similar functions, but the TA group was different (Figure 3B). As shown in Figure 3C, the GHs abundance of the TA group was the highest. However, the GHs abundance of the TBS group showed a higher score than that of the TB group. Furthermore, at the enzyme level, the abundance of α-galactosidase in TBS was the highest, and the abundance of α-galactosidase in TB was also higher as compared to the TA group (Figure 3D). These results suggested that the dominant glycoside hydrolases, including α-galactosidase from rice seed microorganisms, could play a part in the potential interaction effects with rice seeds.

### 2.4. Hybrid Rice Seed Germination after Inoculated with Microorganisms from TA Group, TB Group, and TBS Group

To determine the pathogenicity of the cultivated microorganisms from the chalky rice seeds, we used the cultivated microorganisms of different groups to inoculate the normal hybrid rice seeds during germination. As shown in Figure 4A, the shoot height was the longest in the TA group, and the shoot height in TBS was significantly shorter than that in CK. The germination rate in all groups decreased sequentially from the CK to the TBS group (Figure 4B), and the germination potential and simplified vigor index also showed a similar trend (Figure 4C,D). These agronomic characteristics of seeds during germination indicated that the order of harm to hybrid rice seeds by each microbial group was the TA group, TB group, and TBS group, respectively. The microorganisms from the TBS group were purified from the TB group, and the compositions of those two groups were alike, but the effects caused by the microorganisms from the TBS group were more serious.

## 3. Discussion

The phenomenon of “seed-borne” microorganisms is inevitable in seeds. These seed-associated microorganisms, including bacteria and fungi, survive and propagate by utilizing the seed’s nutritional ingredients. They can proliferate and transmit during their life cycles. Consequently, they are likely to become pathogenic to influence crops during seed storage, germination, development, and growth under specific circumstances [13,14]. Generally, seed-borne microorganisms are divided into four classes: class I is the main source of inoculum where it can be controlled when seed infection is controlled; class II is the pathogens in which seed-borne inoculum is of minor importance; class III is the largest group that never causes infection; class IV is the group that can infect seeds in field or storage, reducing yield and seed quality [15]. From this perspective, for the top-gray chalky grains from hybrid rice seeds, the fourth class of seed-associated microorganisms should be taken into consideration.

The top-gray chalkiness of hybrid rice seeds is a typical phenomenon in hybrid rice seeds. The chalky part of the grain originated from the top side of the endosperm, which is gray and infected by microorganisms. Some factors, including warm weather, moist circumstances during rice heading and filling periods, and the relatively higher glume dehiscence rate in hybrid rice seeds, usually lead to a higher chalkiness rate in seed grains and make seeds more susceptible to infection, resulting in poor seed quality [2,3,4,16]. It had been previously reported that the rice seed-infecting pathogens were fungal microorganisms that could invade hulls and kernels, causing seed quality reduction and seedling diseases [17,18]. In our experiment, bacteria could be cultivated on the beef medium, but they could not grow on the rice flour medium, which was similar to endosperm ingredients. Although the cultivated bacteria could infect the normal seeds during germination, the abundances of bacteria cultivated from the rice flour medium were quite lower. It could be speculated that the bacterial groups were not likely the endophytic seed-infecting sources in the top-gray chalky grains of hybrid rice seeds.

Subsequently, by examining the functions of cultivated fungi, it was found that the genus *Rhizopus* was dominant, and the microbial community owned a high abundance of hydrolytic enzyme genes in this experiment. After subculture, glycoside hydrolases, including α-galactosidase, became dominant. Hydrolytic enzymes are enzymes that use water to break bonds in biomolecules such as peptides, glycosides, and lipids [19]. Glycoside hydrolases hydrolyze oligo- or polysaccharides of carbohydrates to mediate biological functions [20]. A-Galactosidase is an exoglycosidase that decomposes galactoligasaccharides, including raffinose, melibiose, and branched polysaccharides, etc., to remove the terminal α − 1, 6 linked galactose residues [21]. α-Galactosidase can provide nutrition for microorganisms and destroy plant cell walls. Since the main component of rice seed endosperm is starch, it was speculated that the seed-associated fungi in the top-gray chalky grains of hybrid seeds could survive and proliferate by using the polysaccharides in seed endosperm, thereafter, acquire the ability to transmit and infect healthy seeds during storage and soaking.

Koch’s postulates (one pathogen, one disease) are the traditional criteria to determine the pathogenicity of microorganisms [22]. The fungal species in the top-gray chalky grains of hybrid rice seeds are in great numbers, and it is difficult to cultivate these fungi into single strains step by step. In reality, it is quite unlikely that the infection of normal seeds by the top-gray chalky grains during soaking is only caused by one single fungus. To overcome these obstacles and challenges, metagenomic shotgun sequencing does a fair job of figuring out the genotyping species of microorganisms. The established IGC makes it feasible to directly identify genera and even species of microorganisms. Thus, we chose the metagenomic assembly of short sequencing reads to explain the insights of compositions and functions of top-gray chalky grain microorganisms in this experiment, and the microbial cultivation was operated under conditions close to the natural nutrition to avoid the biases caused by isolation and cultivation. The metagenomic shotgun sequencing showed that the hydrolysis ability of microbial communities was enhanced, and the genus *Rhizopus* was dominant. *R. microspores*, *R. delemar*, and *R. oryzae* were dominant in the TB group or the TBS group. *R. microspores*, *R. delemar*, and *R. oryzae* are rich in GHs, and they can make use of polysaccharides of the plant as carbon sources [23,24,25]. The results supported that *R. microspores*, *R. delemar*, and *R. oryzae* were likely to be candidate pathogens in top-gray chalky grains of hybrid rice seeds.

Although it could be speculated that the fungal genus *Rhizopus* was most likely the pathogenic microorganisms responsible for infection during seed storage, transportation, and soaking processes according to its abundance and function, the microorganisms cultivated from the chalky grains were a mixture of fungi and bacteria. Undoubtedly, these microorganisms could synergistically or antagonistically affect normal seeds during the seed germination process, and the seed-borne fungi were the main pathogenic factors [13]. Similar phenomena were also observed in our experiment. For example, the shoot height of the TA group was the longest, although the other germination indexes were lower than CK in the infection validation test. It was probably due to the fact that the from TA group microorganisms were abundant in the genera of *Enterobacter* and *Bacillus* (Figure 2C) and could enhance the shoot growth [26,27].

## 4. Materials and Methods

### 4.1. Materials

Hybrid rice seeds *Indica*-type variety “Longjingyou Dizhan” were provided by Hunan Ava Seeds Co., Ltd., Changsha, China, which were harvested in the summer of 2020 from the Ava Hainan Breeding Experiment Station located in Ledong County, Hainan Province, China (18°29′24″ N, 108°53′24″ E) and were processed in the usual way. Then, the seeds were selected by a YH-101 automatic optical seed chalky rate detector (Hunan Shanzhen Technology Co., Ltd., Changsha, China). The normal seeds and chalky seeds were clarified according to Zhang et al. [6] and Chen et al. [8]. Briefly, the normal seeds were the same seeds selected without chalkiness, and the chalky seeds were mixed from grade 1 to grade 3 using the detector.

### 4.2. Microbial Cultivation

The husks of chalky rice seeds were peeled off in a sterile environment, and then nearly 30 seeds were placed on the Petri dishes with beef extract peptone medium and rice flour medium to cultivate bacteria and fungi, respectively. After incubating at 28 °C for 2 days, the microorganisms cultivated with the rice flour medium were separated by the plating streak method according to Sanders [28] and then incubated at 28 °C for 4 days, and the microorganisms were collected. The samples incubated on beef extract peptone medium without streak were labeled as TA, those incubated on rice flour medium without streak were labeled as TB, and those incubated on rice flour medium after streak were labeled as TBS. TBS was collected 2 days after TA and TB were collected for sequencing. The morphology of fungi was recorded using a Nikon microscope MM-800 (Nikon Co., Tokyo, Japan) with the routine Gram staining procedure [29]. All the samples were stored at −80 °C until further use.

The rice flour medium was prepared with 4.0 g rice flour and 2 g agar in 100 mL water, sterilized, and adjusted to pH 7.2. The beef extract peptone medium was prepared with 0.3 g beef extract, 1 g peptone, 0.5 g NaCl, and 2 g agar in 100 mL water, sterilized, and adjusted to pH 7.2 according to Liu et al. [30].

### 4.3. Metagenomic Shotgun Sequencing and Analysis

Sample DNA was extracted, and qualified DNA was randomly broken into fragments of approximately 350 bp in length [31]. The fragments were modified via terminal repair, adding poly A and sequencing joints, purification, and PCR amplification to build libraries. The libraries from nine samples were sequenced on an Illumina PE150 platform (Illumina Inc., San Diego, CA, USA) at Novogene Biotech Co., Ltd., Beijing, China.

The raw data were pretreated to eliminate low-quality bases and host genome inference to acquire clean data [32]. The metagenome was assembled using MEGAHIT ver1.2.9 [33], open reading frame (ORF) prediction was conducted using MetaGeneMark ver3.25 [34], ORF redundancy was removed using cd-hit ver4.81 [35], and the annotation of species and functions was conducted using DIAMOND ver2.0.2 to blast the unigenes with NR database (ver2018.01) of NCBI [36].

### 4.4. Infection on Seed Germination

Normal seeds of hybrid rice were germinated in plastic growth chambers (17.5 × 12 × 5.5 cm, with chamber lids). The inner bottom of the chambers was padded with two layers of wet pre-sterilized Whatman 1 filter paper. The normal hybrid seeds of three treatments were selected with 200 grains each. After being soaked with 10% hypochlorous acid at room temperature for 10 min, the seeds were washed with sterilized distilled water three times and then were soaked at 30 °C for 24 h to pre-germinate. After that, the seeds were transplanted to the chambers and inoculated at 30 °C and 80% relative humidity with the microorganisms of the TA group, TB group, and TBS group cultivated as above. The microorganisms were inoculated as follows: an inoculum ring of 2.5 mm in diameter was used to evenly coat the microorganisms onto the pre-sterilized Whatman 1 filter paper ten times. Additionally, the treatment without inoculation was used as CK.

The effective germination was confirmed by the following standard: when the length of the radicle equaled that of the seed, the length of the germ was over one-half of the length of the seed [37]. The determination of seed germination indexes, including the germination potential (GP), the germination rate (GR), the simplified vigor index (VI), and the shoot height, was conducted according to Chen et al. and Zhao et al. [8,38]. GP was investigated on the third day after transplantation (d.a.t.), and the other tests were investigated on 7 d.a.t. All of the tests were repeated three times. Briefly, the indexes were calculated according to the following equations:GP = N_3d_/M × 100(1)
where GP is the germination potential, N_3d_ is the number of germinated seeds on 3 d.a.t., and M is the total number of seeds:GR = N_7d_/M × 100(2)
where GR is the germination rate, N_7d_ is the number of germinated seeds on 7 d.a.t., and M is the total number of seeds:VI = GR × H(3)
where VI is the simplified vigor index, and H is the shoot height on 7 d.a.t.

### 4.5. Statistics

All of the tests were performed at least three times unless otherwise stated, the significant differences were calculated using SPSS13.0 (IBM Co., Chicago, IL, USA) with Duncan’s multiple comparison method (*p* < 0.05), and the significant differences were labeled with lowercase letters.

## 5. Conclusions

In this study, we first sequenced the microorganisms of top-gray chalky grains of hybrid rice seeds using metagenomic shotgun sequencing and found that glycoside hydrolases were dominant in the microorganism functions, and *R. microspores*, *R. delemar*, and *R. oryzae* were likely to be the candidate pathogens in top-gray chalky grains of hybrid rice seeds.

## Figures and Tables

**Figure 1 plants-12-02358-f001:**
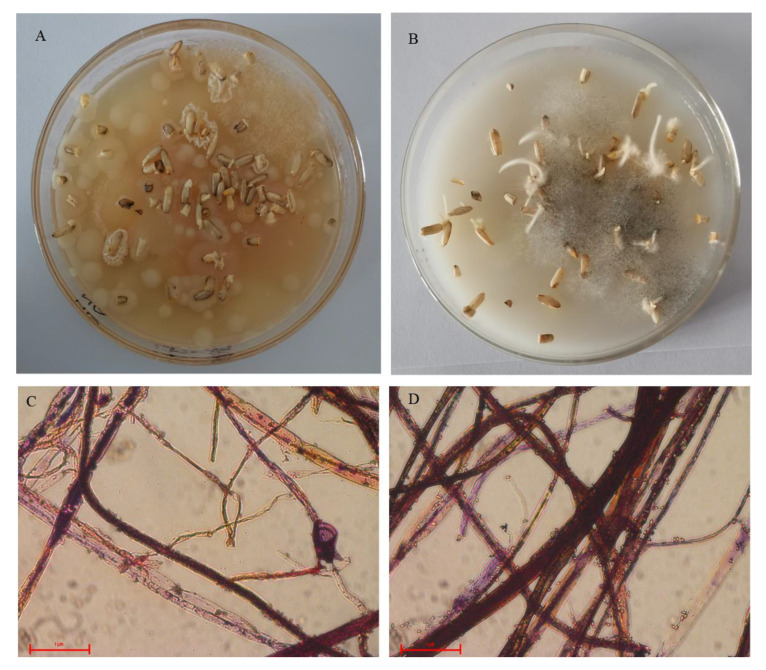
Morphology of microorganisms cultivated from chalky rice seeds. (**A**) Chalky seeds cultivated on beef extract peptone medium; (**B**) chalky seeds cultivated on rice flour medium; (**C**) gram staining (magnification, ×200), cultivated without streak from rice flour medium; (**D**) gram staining (magnification, ×200), cultivated after streak from rice flour medium.

**Figure 2 plants-12-02358-f002:**
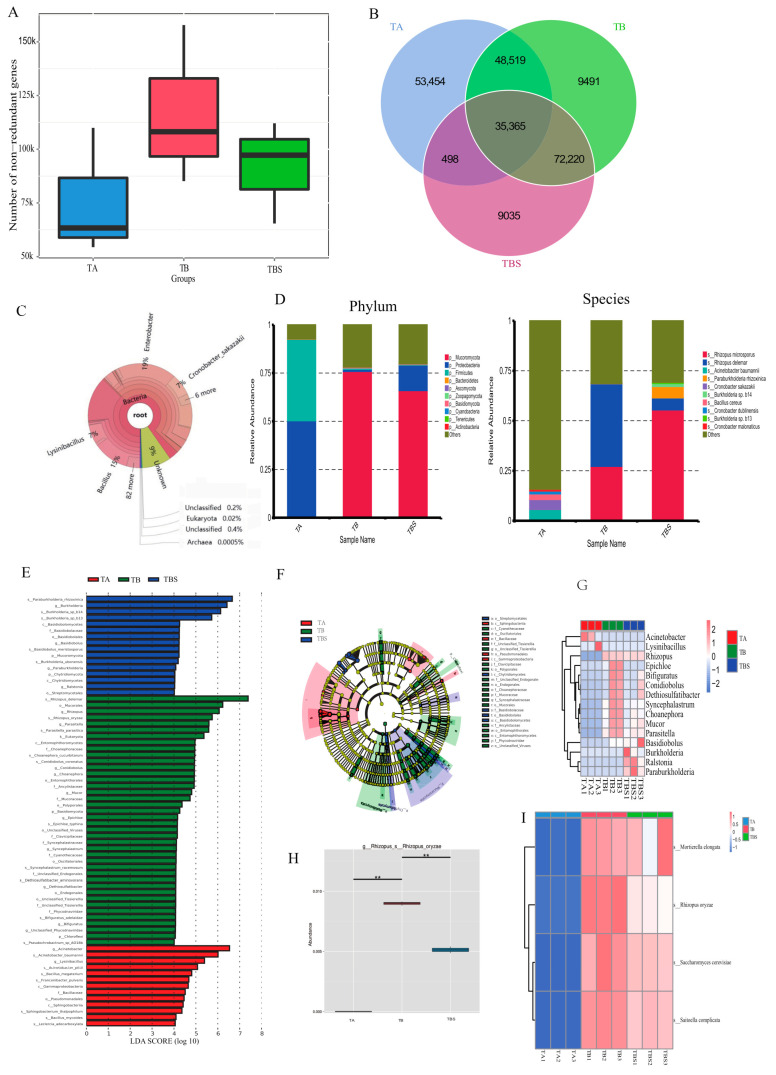
Microbiome diversities in the TA group, TB group, and TBS group based on metagenomic sequencing data. (**A**) Box diagram of gene number differences between groups. (**B**) Venn diagram of observed genes in the TA group, TB group, and TBS group. (**C**) Krona diagram for visualization of relative abundance of taxonomical groups in TA. (**D**) Relative abundance of the TA group, TB group, and TBS group at phylum and species level of top ten members. (**E**) LDA scores for taxa were differentially abundant among TA group, TB group, and TBS group (LDA > 4). (**F**) Cladogram generated by LEfSe indicating differences in taxa among TA group, TB group, and TBS group. (**G**) Heatmap of groups based on differences in taxa by LEfSe. (**H**) Boxplot generated by Metastats indicating significantly different species among TA group, TB group, and TBS group. ** represented highly significant differences between groups (q < 0.01). (**I**) Heatmap of species with significant differences by Metastats indicating significantly different species among TA group, TB group, and TBS group.

**Figure 3 plants-12-02358-f003:**
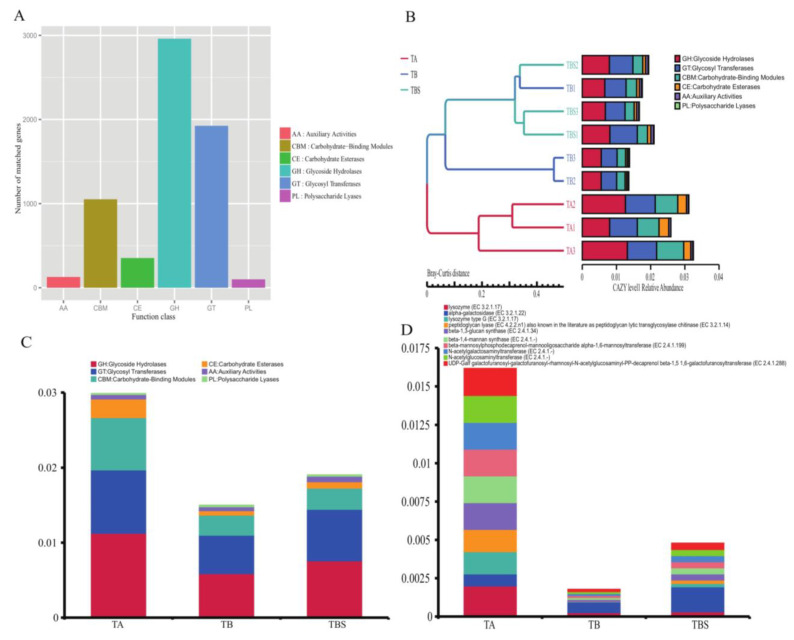
Function diversities in the TA group, TB group, and TBS group based on metagenomic sequencing data. (**A**) Relative abundance of annotated gene categories at level 1. (**B**) Gene clusters of samples at level 1 by Bray–Curtis distance matrix. (**C**) Relative abundance of annotated gene categories at level 1 among groups. (**D**) Relative abundance of annotated gene categories at the top 10 enzyme level among groups.

**Figure 4 plants-12-02358-f004:**
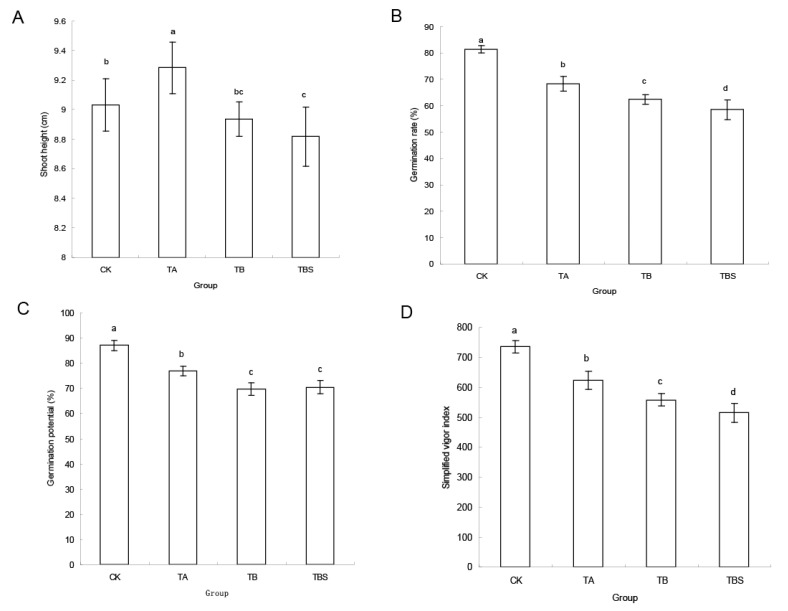
Germination indexes of normal hybrid rice seeds after inoculation with microorganisms of the TA group, TB group, and TBS group. Lowercase letters stand for significant difference. (**A**) Shoot height. (**B**) Germination rate. (**C**) Germination potential. (**D**) Simplified vigor index.

## Data Availability

Illumina-Hiseq-generated metagenomic reads are available in the NCBI Sequence Read Archive (SRA) for the Bioproject: PRJNA974197.

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
