# Peer review of "Metagenomic Sequencing to Analyze Composition and Function of Top-Gray Chalky Grain Microorganisms from Hybrid Rice Seeds"

_plants, 2023, doi:10.3390/plants12122358_

Round 1
Reviewer 1 Report (Previous Reviewer 1)
The relevance of the answers proposed by the authors was assessed in relation to the questions previously posed in the evaluation of the manuscript. In general, the authors have provided accurate and appropriate answers to the various questions. In addition, the authors have made useful corrections and provided additional information with supporting references in paragraph form, particularly in the introduction and discussion. This has undoubtedly improved the scientific quality of the new version of the manuscript.
I suggest that the authors delete and reposition in the correct place the reference [14] in the section Result/ 2.1. Microorganisms grown from chalky rice seeds.
In view of the above, I give my consent for the article to be published as is in your journal.
Author Response
Dear Dr.,
Thank you very much for your suggestion. We deleted reference 14 in 2.1, and we added reference 17: Ackaah et al. Int. J. Clin. Microbiol., 2023, 2023, 8690464 in section Discussion. And the reference sequence was adjusted accordingly. All of the changed in the manuscript were in MS word revision mode.
Best Regards,
Yihong Hu
College of Agriculture and Biotechnology, Hunan University of Humanities, Science and Technology
Reviewer 2 Report (Previous Reviewer 2)
The authors modified accordingly
The authors modified accordingly
Author Response
Dear Dr.,
Thank you very much for your work. We invited a colleague who is fluent in English to help us revise the manuscript. We revised the manuscript carefully. The following changed were made in the manuscript:
- In the third to last row of Abstract, we deleted an abundant “in”.
- The first and second sentences of Introduction were changed as follows:
Hybrid rice (Oryza sativa L.) is one of the main staple foods for people in the world and China is a major producing area for hybrid rice. Since the 1970s, seed production technology of hybrid rice has developed rapidly.
- In the last sentence of the first paragraph of Introduction, “reducing” and “polluting” were used instead of “reduce” and “pollute”, respectively.
- In the second sentence of the second paragraph of Introduction, “impacting” was used instead of “to impact”.
- In the first sentence of the third paragraph of Introduction, “approach” was used instead of “policy”.
- In the second sentence of the fourth paragraph of Introduction, “But” was deleted.
- The first sentence of 2.1 was changed into “From the chalky rice seeds without husks, microorganisms were cultivated on beef extract peptone medium (TA) and rice flour medium (TB), respectively.”
- In the second sentence of 2.4, “that” was inserted.
- We revised the second sentence of the first paragraph of Discussion.
- In the last sentence of the third paragraph of Discussion, an abundant “and” was deleted.
- The last sentence of Discussion was rewritten.
All of the changed in the manuscript were in MS word revision mode.
Best Regards,
Yihong Hu
College of Agriculture and Biotechnology, Hunan University of Humanities, Science and Technology
This manuscript is a resubmission of an earlier submission. The following is a list of the peer review reports and author responses from that submission.
Round 1
Reviewer 1 Report
see attached

Reviewer 2 Report
Abstract: here, it is not clear the provenience of microorganisms sequenced. The authors fist should talk about which kind of microorganisms they are talking about (inoculants, infectants, endophytes, seed-associated....)
Introduction: The author should deep the issue of importance of microbiota associated to seeds.
Material and Methods: d must be "days".
It is not clear the differentiation of TA, TB and TBS. Explain why the authors differentiate these treatement plating in this section.
How can authors "interrupted into about 350 bp fragments in length" ? Then, add refs for library protocol preparation.
Results: line 73: to see what? Repeat here.
Reading the manuscript, it is highlighted the footprint of the paper: the topic and the data reported are more close to a microbiological work than a "Plant" focus work.
The results report data regarding microbes colonization of seeds and.the taxonomical composition of seed-native microorganisms.
To be improved